# Internet Use and the Happiness of Rural Residents: The Role of Education and Health

**DOI:** 10.3390/ijerph20043540

**Published:** 2023-02-17

**Authors:** Yan Mei, Nuoyan Lin

**Affiliations:** School of Economics, Hangzhou Dianzi University, Hangzhou 310000, China

**Keywords:** education, happiness, health, human capital, mediation effect

## Abstract

The rapid development of the Internet in China in recent years has greatly penetrated into all aspects of people’s lives and production. In rural areas of China, little is known from previous studies about the relationship between the Internet and happiness. Based on data from the China Family Panel Studies (CFPS) collected in 2016 and 2018, this study investigates the impact of the Internet use on the happiness of rural residents and its mechanism. The results show that, first, the fixed-effects model indicates that the Internet significantly increases the happiness of rural residents. Second, the multiple mediating effects analysis shows that Internet use can improve rural residents’ happiness by promoting the household education human capital. To be more specific, excessive Internet use leads to lower levels of household health human capital. However, a lower level of health does not necessarily bring about a lower level of happiness. In this paper, the mediating effects of household education human capital and household health human capital are 17.8% and 9.5%, respectively. Third, the heterogeneity analysis found that there was a significant positive correlation between Internet use and rural residents’ happiness in western regions of China, while it is insignificant in the eastern and central regions; for households with a huge labor force, Internet use dramatically improves their happiness by enhancing their household education human capital. Education and health have different roles to play in terms of the happiness of rural residents. Therefore, this suggests that the physical and psychological health of rural residents should be taken into account when the Internet strategies to improve general well-being are under development.

## 1. Introduction

With the advent of the new technological revolution, the world has entered an information age, and the Internet plays an increasingly important role as a major information medium in promoting economic and social development. Smartphones, computers and other devices people possess, as well as local areas of Internet access, can increase farmers’ income and expenses and, finally, improve economic well-being [1,2], support for rural economic and technological development [3], improve agricultural productivity [4], promote farmers’ market participation and reduce search costs [5], as well as facilitate the use of mobile money, thereby increasing savings and improving the welfare of smallholder farmers [6,7]. In 1994, China was connected to the Internet with a 64 K international dedicated line, achieving a fully functional connection to the international Internet. According to the 50th Statistical Report on the Development Status of the Internet in China released by China Internet Network Information Center (CNNIC), as of June 2022, the size of rural Internet users in China reached 293 million, and the Internet penetration rate in rural areas was 58.8%. It can be seen that, in China, the Internet has helped revitalize the countryside, and the digital transformation of rural areas has continued to deepen.

In May 2019, the General Office of the Central Committee of the Communist Party of China issued the Outline of the Digital Rural Development Strategy, which aims to continuously generate endogenous momentum for rural development and continuously improve farmers’ sense of gain, happiness and security. The emergence of the Internet has brought unprecedented changes to all aspects of rural residents’ life and production and has also had a great impact on their sense of happiness. In response to this phenomenon, some studies have concluded that the reason why Internet use can increase happiness is mainly due to its ability to offer more life satisfaction, better access to information and enhanced social interaction [8,9,10]. However, some scholars have pointed out that the use of the Internet has increased social isolation and has decreased social trust [11,12,13]. In either case, the Internet and household human capital is closely connected. Thus, as important components of human capital, education and health will have a significant impact on the happiness of rural residents. Judging from the current situation, the two-week prevalence rate of rural residents in China increased from 20.2% in 2013 to 32.2% in 2018, which means that the overall health status of rural residents in China is on a decreasing trend.

The “Easterlin paradox” states that a better economy and higher per capita incomes do not necessarily lead to a corresponding increase in gross national happiness [14]. Oswald’s (1997) [15] study also supported that a significant increase in per capita incomes does not improve people’s happiness. Subsequently, many scholars began to work on relevant research. What is happiness? For decades, there have been varying degrees of disagreement in academia. Early philosophers defined happiness in various ways. For instance, Tatarkiewicz (1976) [16] equated happiness with success, and Socrates associated happiness with wisdom. The sensualist view of happiness, represented by Jeremy Bentham, holds that sensuality is the source of a happy life and constitutes the basis of happiness and morality. By the mid-20th century, psychologists tried to give their answers with the help of science. They have construed happiness as the evaluator’s holistic assessment of the quality of his or her life based on self-determined criteria [17], a definition that focuses on the individual’s subjective assessment of life, putting forward the concept of subjective well-being. Ryff (1989) [18] further classified subjective well-being into three categories: “external criteria” based on the observer’s perspective, “internal emotions” and “personal self-evaluation” from the individual perspective. Another psychologist, Diener, conducted an in-depth study of subjective well-being, supposing that happiness is a comprehensive evaluation with three characteristics: subjectivity, relative stability and wholeness [19]. Additionally, based on the research on subjective well-being in the past three decades, he divided subjective well-being into four dimensions: life satisfaction, positive emotion, negative emotion and satisfaction with all aspects of life [20]. Whereas positive emotions regulate the individual psyche and guarantee that the individual achieves higher subjective well-being, negative emotions do the opposite [21]. Since then, research on subjective well-being has entered a new stage.

Based on multidisciplinary perspectives, the literature has been richly researched on the connotation of happiness and its influencing factors, respectively, resulting in a large number of drawable research results. There are numerous factors that can affect happiness, and studies in different geographical, social and economic contexts often present different results. Current research on the subjective well-being of Chinese rural residents has some limitations, such as simple research methods, inadequate research data, insufficient research factors, etc. Therefore, this paper empirically investigates the intrinsic link between Internet use and the happiness of Chinese rural residents based on the 2016 and 2018 China Family Panel Studies (CFPS), explores the transmission mechanism behind them and analyzes its heterogeneity in order to answer the following questions:Does Internet use promote the happiness of rural residents?What are its mechanisms of action?What are the differences in the effects of rural residents among different regions and household structures?

This paper hypothesizes Internet use improves rural residents’ happiness, and Internet use improves household education human capital, thus it increases the happiness of rural residents; however, Internet use deteriorates household health human capital. At the same time, rural residents with poorer household health human capital do not have a concomitant decrease in happiness.

The marginal contributions of this paper are as follows. First, this paper explores the mediating role of education and health between the Internet and happiness, providing new empirical evidence for the relationship between Internet use and the happiness of rural residents and expanding the relevant research on the economics of happiness. Second, this paper further explores the heterogeneous effects of different regions and different household structures on the happiness of rural residents. In addition to classifying regions by eastern, central and western regions, household structure is differentiated by the number of household laborers to provide more empirical evidence for policy research.

## 2. Literature Review

Numerous scholars have conducted various research on the connotation of subjective well-being. Recently, scientists have focused more on what causes people to subjectively feel that their lives are worthwhile and meaningful instead of what defines a good life [22]. A considerable amount of literature has focused on the impact of income on well-being, such as income distribution, relative income and absolute income [23,24,25]. Among them, Easterlin’s study found a positive relationship between income and happiness, but the increase in happiness does not exactly coincide with income growth [26]. Smyth and Qian (2008) [27] pointed out that, setting average income as a cut-off, the increase in happiness from relative income was greater than the increase in average income for those earning more than average income; conversely, the increase in happiness from relative income was less than the increase in average income. Some literature has focused on the impact of personal characteristics such as age, gender, education and health on happiness [28,29,30,31]. Still others focused on the impact that some social characteristics have on happiness, such as unemployment, inflation and climatic conditions. For example, some scholars found that the sample reported a strong correlation between happiness and inflation and unemployment rates through a panel data analysis of countries [32]. Evidence of the effects of pollution and environmental factors on happiness was very limited in the early days [33]. The earliest evidence was from Welsch (2002) [34], who used large air pollution data from 54 countries to show that air pollution significantly worsens happiness when using nitrogen dioxide emissions as an indicator. Additionally, extreme weather could have a negative impact on happiness [35].

In recent years, many studies have concerned the impact of Internet use on people’s happiness. It could be found that the conclusions are various due to differences in data, samples and empirical methods. On the one hand, some studies have found a possible negative impact of Internet use on happiness. Kraut et al. (1998) [12] noted that the Internet can lead to less family communication and smaller social circles. By meta-analysis and 40 existing studies, Huang (2010) [36] found that Internet use impairs happiness, which includes feelings of loss and loneliness, low self-esteem and a lack of life satisfaction. Additionally, for those who have already felt lonely, time spent on social networks is negatively associated with levels of happiness [37]. Additionally, as the most popular activity on the Internet, social media use is associated with lower task performance, increased technological stress and less happiness [38]. In addition, problematic Internet use has a negative impact on happiness [11]. For example, using the Internet for prank-related activities is associated with lower levels of well-being and social support [39]. The proliferation and easy availability of information are also parts of its negative impact. People who frequently access information via the Internet experience lower levels of satisfaction from their income [40]. Similarly, if this information is used and compared excessively, it can lead to a decrease in people’s offline communication, resulting in residents less actively participating in social activities [9].

On the other hand, some studies have found that the Internet, as a new tool for informatization and digitization, can promote happiness. Non-Internet users are less satisfied with their lives than Internet users, and Internet use has a greater positive impact on young people or on individuals who are dissatisfied with their income [13]. Valenzuela et al. (2009) [10] noted that the intensity of Facebook use is positively correlated with life satisfaction and social trust. Castellacci and Viñas-Bardolet (2019) [41] argued that Internet technology increases job satisfaction by improving access to data and information, creating new activities and facilitating communication and social interaction. In addition, some occupations, as well as workers with higher income and education levels, benefit relatively more from the Internet than those with weaker ties to ICT activities. The Internet can improve efficiency, effectively reduce time costs [42] and provide different products and services to create a new model of daily life [43]. Some scholars have also explored the link between the Internet and happiness based on specific behaviors. Social network sites benefit users when they are used to build meaningful social relationships, otherwise, with traps such as isolation and social comparison, social network sites can be detrimental to users [44]. Proper use of social network sites can boost happiness by creating social capital and stimulating social connection [45].

The above studies are the results of various scholars based on certain countries and subjects, which can be a reference to rural areas of China. Achieving better living conditions is one of the main goals of the government today [9]. In the background of the era of full penetration of the Internet into rural areas, this paper proposes the first research hypothesis.

**Hypothesis** **1** **(H1).***Internet use improves rural residents’ happiness*.

In recent years, a large amount of literature has studied the direct link between happiness and education. Some literature suggests a positive correlation between education and happiness [46,47,48]. Cuñado and de Gracia (2012) [30] analyzed the direct and indirect contribution of the level of education to happiness: the acquisition of knowledge facilitates the improvement of self-confidence, self-evaluation and pleasure; education promotes better employment, higher quality of work and job salary. In addition, education enables people to adopt better lifestyle habits such as exercise habits, eating habits and drinking habits [48]. Sulemana et al. (2017) [49] note that education raises income levels and improves people’s standard of living, thereby improving their happiness. Other literature suggests that the specific correlation between education and happiness is not clear. Inglehart and Klingemann (2000) [50] noted that education has no significant effect on life satisfaction. Clark and Oswald (1996) [51] argued that the higher the job expectations of educated people, the wider the income gap as education increases. As a result, education actually decreases satisfaction.

In the long run, the increase in education level significantly enhances the welfare of Chinese residents [52], so this paper proposes the following hypothesis.

**Hypothesis** **2** **(H2).***Internet use enhances rural residents’ household education human capital*.

**Hypothesis** **3** **(H3).***Internet use improves household education human capital, thus increasing the happiness of rural residents*.

Several studies have found that Internet use positively and significantly affects health outcomes through increased access to information, social interaction and physical ability [53]. Korp (2006) [54] emphasized the need to assess the sources of health information on the Internet and noted that the Internet accelerates the dissemination of health knowledge and is more conducive to health promotion. Specifically, health information provided by the Internet can cultivate health-related behaviors and improve personal health literacy [55], promote health equality, raise health awareness and reduce depression [56]. The use of the Internet can provide and improve information services in health care [57], break the monopoly of doctors on professional information, alleviate the information asymmetry between doctors and patients and help residents better manage their health [58]. It may change the traditional structure of the doctor–patient relationship and help patients better understand their doctor’s recommendations [59].

However, some literature suggests that the Internet can also have negative effects. In the area of mental health, the Internet provides individuals with access to a wealth of information about health, but the credibility of this information is highly uncertain. Access to information via the Internet may promote medicalization and healthism, making the issues in daily life pathological [60]. Unregulated Internet use can lead to increased emotional fatigue, which translates into poor physical performance and perceptions, such as increased anxiety and depression [61,62]. Ho et al. (2014) [63] also show that extreme Internet use can lead to psychological problems such as anxiety and depression. In terms of social interaction, excessive use of the Internet for communication and access to information can contribute to a decrease in the quality of life of older adults [64]. Bessière et al. (2010) [65] noted that health-related Internet use was associated with a small but reliable increase in depression. Lin assessed depression and social media use on multiple social media platforms based on a large sample of youth, finding a significant positive association between social media use and depression. Additionally, this association was shown to be significantly correlated with the number of different social media platforms used [66]. In terms of physical activity, Internet use can lead to sedentarism and obesity, potentially negatively impacting health outcomes [67,68]. The above-mentioned studies suggest that Internet use may lead to a decrease in health. Health may be an important mediator of the relationship between the Internet and happiness. As a result, H4 and H5 are proposed.

**Hypothesis** **4** **(H4).***Internet use deteriorates household health human capital*.

**Hypothesis** **5** **(H5).***Rural residents with poorer household health human capital do not have a concomitant decrease in happiness*.

## 3. Materials and Methods

### 3.1. Data Sources

The data in this paper are from the China Family Panel Studies (CFPS), which was implemented by the Institute of Social Science Survey (ISSS) and funded by the 985 Project of Peking University. In 2010, CFPS launched a formal nationwide survey with a size of 16,000 households and an overall target coverage of rural and urban areas in 25 provinces/municipalities/autonomous regions in China, originally planned to be conducted once a year (later changed to once every two years); six rounds of national surveys have been conducted, including 2010, 2012, 2014, 2016, 2018 and 2020. The data used in this paper are CFPS2016 and CFPS2018, and considering that the rural revitalization strategy was formally proposed in 2018, the use of data from the two periods before and after can reflect the rapid development of rural digitalization and the effectiveness of policy implementation during this period. In this paper, we screen out residents with rural households, with a valid sample of 9482 for the two-period balanced panel data and a valid sample of 3322 for the 2018 cross-sectional data. We use Stata11 software for data matching and integration to match the household economic database, as well as the adult database by individual code (PID), household sample code (FID18) and financial respondent (RESP1PID). Since the CFPS2018 does not define the head of household, the “financial respondent (RESPIPID)” is used as the head of household in this paper, and the missing values in both periods are directly excluded. In addition, according to the research needs of this paper, the respondents who lived in a community type of village committee are defined as farming households. The samples in which the community nature (FA1 and FA101) is “village committee” are retained, and the samples in which the questionnaire options are “do not know” or “refused” to answer are excluded.

### 3.2. Selection of Variables

#### 3.2.1. Happiness Measurement

Scholars often use respondent-reported happiness scores or happiness ranking assignments to examine a person’s happiness. In this paper, the happiness of rural residents is used as the explanatory variable and the respondent-reported self-scores from 0 to 10 are used as a measure of their happiness. Since most data of the happiness reported in the 2016 sample are missing, the 2016 data are replaced by respondents’ self-rated life satisfaction.

#### 3.2.2. Internet Use Measurement

In order to examine the impact of Internet use on the happiness of rural residents, we combine theoretical analysis with the length of Internet use as the core explanatory variable, measured by the CFPS2018 question “how many hours per week are spent on the Internet”, and the data obtained are divided by seven and multiplied by sixty and logged to examine the average daily length of Internet use.

#### 3.2.3. Mediating Variable

“Household human capital” is the mediating variable in this paper, while health and education are selected as its measures.

Household education human capital. Measured by years of education in the household workforce. Years of education of household members are captured by CFPS, and the mean value of years of education in the sample household workforce is calculated.Household health human capital. Measured by the mean value of family workforce health self-assessment. The CFPS investigates the self-rated health status of household members in five categories: “extremely healthy”, “very healthy”, “relatively healthy”, “general healthy” and “unhealthy”, and then calculates the mean health status of the household workforce.

#### 3.2.4. Control Variables

The control variables are personal characteristics, household characteristics and regional characteristics in the analysis process. The personal characteristics include gender, age, education level, health status and political situation; the household characteristics variables mainly include household size and per capita household income; and the regional characteristics variables mainly include the province and social status of rural residents. Considering that the degree of development and Internet penetration vary in different regions, dummy variables are used to control for regions.

### 3.3. Research Methods

#### 3.3.1. Regression Model

To test whether Internet use has a role to play in differed levels of the happiness of rural residents, the regression model in this paper is as follows:(1)$${\mathrm{Happiness}}_{i,j}=\alpha_{1}{\mathrm{Itud}}_{i,j}+\delta_{1}X_{i,j}+\mu_{t}+\varepsilon_{i,j}$$

The subscripts i and j indicate the individual and the period, respectively. The explanatory variable ${\mathrm{Happiness}}_{i,j}$ indicates the level of happiness of individual i in period j. The explanatory variable ${\mathrm{Itud}}_{i,j}$ denotes the length of Internet usage of individual i in period j, $X_{i,j}$ indicates the control variables for individual characteristics, family characteristics and regional characteristics that play a part in differing the level of happiness of individual i, $\mu_{t}$ indicates area fixed effects and $\varepsilon_{i,j}$ indicates the random error term.

#### 3.3.2. Mediation Model

In this paper, based on the research methodology proposed by Wen and Ye (2014) [69], the mediating effect model is tested using Mplus, and the mediating effect model is as follows:(2)$${\mathrm{HC}}_{i,j}=\alpha_{2}{\mathrm{Itud}}_{i,j}+\delta_{2}X_{i,j}+\varepsilon_{i,j}$$
(3)$${\mathrm{Happiness}}_{i,j}=\alpha_{3}{\mathrm{Itud}}_{i,j}+{\beta \mathrm{HC}}_{i,j}+\delta_{3}X_{i,j}+\varepsilon_{i,j}$$

${\mathrm{HC}}_{i,j}$ denotes the household human capital, including household education human capital and household health human capital. Firstly, the coefficient $\alpha_{2}$ in Equation (2) is tested against the coefficient β in Equation (3), secondly, the coefficient $\alpha_{3}$ in Equation (3) is tested, and finally, 5000 samples are drawn using the bootstrap method to estimate the magnitude of the mediating effect $\alpha_{2}\beta$.

## 4. Results

### 4.1. Baseline Model

#### 4.1.1. Descriptive Statistics

Table 1 shows that the overall mean value of happiness is 5.550, which means that the overall happiness is at a moderate to high level. In terms of personal characteristics, the proportion of men and women in the sample is basically equal, with slightly fewer women than men; the average value of years of education is 10.45, indicating that the sample is mostly in the state of not having graduated from high school; the mean value of health status is 2.618, which is lower than the median value of 3, indicating that the overall self-rated health status of the sample is below the medium level. The number of party members accounts for 7.4% of the full sample, with nonparty members constituting the majority. In terms of family characteristics, there are basically more than four family members; in terms of social characteristics, the mean value of the sample’s social status self-rating is 2.8, which is below the medium level, and it can be seen that the sample generally considers itself to be of low social status.

#### 4.1.2. Baseline Regression Result

Table 2 reports the regression results of the relationship between Internet use and the happiness of rural residents. Models (1) and (2) control for variables of individual characteristics only, and for models (3) and (4), models (5) and (6) add control variables of household characteristics and regional characteristics to the former in turn. Fe and Re denote the fixed-effects model and the random-effects model, respectively. This table shows that regardless of the inclusion of control variables, the Internet use and happiness of rural residents are significantly positively correlated, and hypothesis 1 is tested. Taking model (5) as an example, for each unit increases in Internet use, the happiness of rural residents increases by 0.063 unit, and we can assume that Internet use has a positive contribution to the happiness of rural residents. On the one hand, the construction of rural information infrastructure has accelerated in recent years, the penetration rate of cell phones and computers has steadily increased and more and more farmers acquiring information and broadening their horizons through mobile devices, computers, etc., has gradually become the norm. On the other hand, the Internet has empowered farmers’ ways of life, and the network has been expanding in rural application scenarios. It is concentrated in all aspects of rural life, such as medical treatment, shopping and broadband Internet access. Based on the “Internet+” strategy in China, the majority of farmers are able to enjoy good living convenience like urban residents, and their sense of happiness has increased.

The results from other groups with control variables also present differently on the happiness of rural residents. Among the personal characteristics, the regression coefficient of age is 53.978, which means that rural residents who are older have a higher sense of happiness. In contrast, the health status of rural residents has a negative correlation with their happiness. The possible reason for this is that the increasingly fast pace of life has led them to sacrifice job satisfaction and increase work intensity, resulting in poorer health, but at the same time, income has increased and happiness has improved. Among family characteristics, individuals with a larger number of family members, as well as a better per capita household income status, enjoy higher well-being. Among others, individuals with higher self-ratings of their local social status also have higher levels of happiness.

### 4.2. Mediation Model

Table 3 shows the results of the mediating effects model based on CFPS2018 with household human capital as a mediating variable. When we examine both the direct relationship between Internet use and happiness and the indirect relationship between Internet use, household human capital and happiness, both the direct and indirect effects pass the significance test. That is, there is a partial mediating correlation between household human capital and Internet use on the happiness of rural residents.

First, Internet use has a positive correlation with household education human capital (β = 0.503, *p* < 0.001); household education human capital enhances the happiness of rural residents (β = 0.038, *p* < 0.05). This means that Internet use can effectively improve the level of household education human capital, and household education human capital in turn promotes the happiness of rural residents, so that hypothesis 2 and hypothesis 3 are verified.

Second, Internet use has a significant negative relationship with household health human capital (β = −0.032, *p* < 0.05). Yet lower levels of health do not mean that the sample reports poorer levels of happiness (β = −0.32, *p* < 0.001). It means that Internet use reduces household health status, while rural residents’ happiness does not decrease, which verifies hypotheses 4 and 5. Considering the data in this section are survey data from 2018, while most of the previous studies used data prior to 2015, a possible explanation is that in the early days when the Internet was not so widespread, rural residents had limited access to the Internet due to space and time constraints. In recent years, the gap between urban and rural access to the Internet has gradually narrowed, and rural groups have accelerated their integration into the online society. Although excessive use of the Internet leads to a decline in health, at the expense of health, the increase in income due to Internet applications in turn provides farmers with psychological satisfaction and a corresponding increase in happiness. Figure 1 shows a visualization of the mediating effect.

According to Table 4, it can be seen that the test of mediating effect based on Bootstrap method finds that the mediating effect (1) of household education human capital between farmers’ happiness and Internet use is 0.019. Its deviation-corrected bootstrap confidence interval at the 95% confidence level is [0.007, 0.033] and does not contain zero. This indicates the existence of a mediating effect of household education human capital on Internet use and happiness, with the mediating effect accounting for 17.8% of the total effect. The size of the mediating effect (2) of household health human capital between happiness and Internet use among rural residents is 0.010. Its deviation-corrected bootstrap confidence interval at the 95% confidence level is [0.002, 0.021], which does not contain zero. It indicates the existence of a mediating effect of household health human capital on Internet use and happiness, with the mediating effect accounting for 9.5% of the total effect.

### 4.3. Heterogeneous Analyses

#### 4.3.1. Regional Heterogeneity

The above analyzed the mediating effect of household human capital between Internet use and the happiness of rural residents. However, in fact, due to the limitations of economic development and geography in rural areas and differences of human capital and social capital among different groups, people have very different initial motivations and usage patterns of Internet use. There may be differences in the mediating effects of household human capital, so the heterogeneity is further examined by region and household structure.

According to Table 5, from the perspective of different regions, no matter in eastern, central or western China, Internet use and rural residents’ household education human capital are significantly positively correlated, and household health human capital is negatively correlated with rural residents’ happiness. Moreover, only in the western region, Internet use still significantly increases the happiness of rural residents after the inclusion of the mediating variable (household human capital) (β = 0.159, *p* < 0.01). It may be that, in the past two years, the Internet penetration in the eastern and central regions has been higher, so the impact of the Internet is smaller, while in the west, because of the backward development, the dividends from the Internet penetration have been more significant.

#### 4.3.2. Household Structure Heterogeneity

There are three groups according to the number of household laborers: (1) number of household laborers < 2; (2) number of household laborers = 2; and (3) number of household laborers > 2. According to Table 6, from the perspective of household structure, the correlation between Internet use and happiness is no longer significant after the inclusion of mediating variables in model (1). At this time, the direct effect is not significant, only the mediating effect is. For households with a small family workforce, both household education human capital and household health human capital contribute to the happiness of rural residents. Both paths in model (2) are not significant. The mediating effect of household health human capital in model (3) is insignificant, while the mediating effect of household education human capital is significant. Thus, for rural households with a huge labor force, Internet use would significantly contribute to their happiness by improving the level of education human capital.

## 5. Discussion

Based on panel data from CFPS and cross-sectional data, this paper analyzes the relationship between Internet use and the happiness of Chinese rural residents by constructing a panel model as well as a mediation model. Additionally, by introducing household education human capital and household health human capital as mediating variables, this paper explores the transmission mechanism of rural residents’ happiness. Multiple mediator analysis is conducted by using Mplus software to understand the relationship between Internet use and happiness. The main conclusions are shown in Figure 2.

Firstly, H1 is verified. Consistent with previous studies, the results of this paper’s benchmark regression find that, after controlling for individual characteristics, household characteristics and regional characteristics, both the fixed-effects model and the random-effects model indicate that Internet use and the happiness of rural residents have a significant positive relationship [8,13]. This result also supports other scholars’ earlier arguments that the Internet is beneficial in widening access to information and data, improving efficiency, saving time, facilitating communication and social interaction, etc., and thus improving people’s happiness [41,42].

Secondly, H2 and H3 are verified. The multiple mediation model analysis showed that there was a partial mediation effect of household human capital in the relationship between Internet use and rural residents’ happiness. Each unit increase in Internet use leads to a 1.9% increase in rural residents’ happiness through household education human capital. It implies that Internet use can contribute to the happiness of rural residents by improving household education human capital. This is generally consistent with the existing literature, which shows that the increase in education level significantly improves the welfare of Chinese residents [52]. In contrast, the study by Clark and Oswald (1996) [51] showed the opposite: satisfaction and education level were negatively correlated. The possible reason is that “human psychology is partly determined directly by the economy and partly by the whole political system on which it grows” (Plekhanov, 1908) [70]. In China’s rural areas, with the increasing improvement of income and living quality, people generally have the mentality of “be content with little wealth”. As a result, they do not have higher job expectations and are more likely to feel satisfied.

Finally, H4 and H5 are verified. Internet use increased by one unit, while household health human capital decreased by 3.2%. Instead, when household’s human capital decreases by one unit, happiness increases by 32%. This implies that Internet use worsens household health human capital without reducing happiness. Possible explanations are that while Internet use correlates with poorer mental health, lower physical activity levels and negatively correlates with health outcomes [64,68], people with poor health are not relatively unhappy, because they may have adapted to their physical condition, which is known as the “disability paradox” [31]. In addition, the heterogeneity analysis finds that the positive correlation between Internet and happiness is more significant in western regions. For rural households with a large labor force, the mediating role of education human capital is significantly higher than that of health human capital.

Existing studies do not agree on whether Internet use can enhance happiness and health status. Therefore, a threshold model can be introduced in the future to explore the relationship of Internet use to happiness and health status in stages. Thus, the association between Internet use, subjective well-being and health status can be revealed more comprehensively. Secondly, there are different content types of Internet use, and future research can divide Internet use into several dimensions to explore the relationship between internet use and rural residents’ subjective well-being through human capital. Eventually, some of the variable measures in this paper are respondents’ self-assessments, which may be subject to some bias. Future research may need to expand on and refine this issue.

In addition, during the pandemic, people became more vulnerable to loneliness and anxiety due to spatial constraints. The use of digital technology can alleviate stress and boredom, but excessive use of social media may worsen people’s mental problems and have negative effects [71]. Problematic Internet use has also increased dramatically as a result of the pandemic, particularly among young people and those who were infected [72,73]. Some vulnerable groups do not use the Internet to access information or socialize and are less likely to benefit from it than those who do [74]. Therefore, future research should focus on the impact of Internet use on happiness during the pandemic and how it differs among different populations. Thus, the rational use of the Internet in the context of a major global health crisis needs to be researched urgently.

## 6. Conclusions

The positive correlation of Internet use on the happiness of Chinese rural residents indicates the importance of Internet strategies for rural development policies and urban-rural integration. First, given the significant positive impact of Internet use on the well-being of rural residents, the government should continue to improve Internet infrastructure, narrow the digital divide between urban and rural areas and provide convenient, fast and high-quality Internet services so that the general public can share the benefits of development in the Internet era. Second, online education needs to be developed to continuously raise the level of education human capital of the labor force in households and to expand the stock of education human capital. At the same time, the government should pay more attention to the underdeveloped rural areas. The digital interface of the Internet can help to provide various learning resources and facilitate rural residents’ access to education resources, thus increasing the size and proportion of the population receiving higher education and promoting the improvement of rural residents’ subjective well-being. Third, the government should actively regulate Internet health information platforms to reduce rural residents’ exposure to false health information. Moreover, authoritative information and knowledge should be provided for rural residents to prevent and resolve the security risks of Internet health services. At the same time, the government should gradually carry out targeted Internet health education, explore diversified forms of Internet medical services and improve the health status of rural residents. Fourth, the government should keep promoting the balanced development of household education human capital and health human capital between urban and rural areas, promote the integrated development of urban and rural areas and ensure that basic public services are shared by all. Additionally, the government should improve the mechanism for the balanced allocation of urban and rural education and medical resources, actively develop rural education, implement the digital rural development strategy, fully build a digital village and finally help realize an overall rural revitalization.

## Figures and Tables

**Figure 1 ijerph-20-03540-f001:**
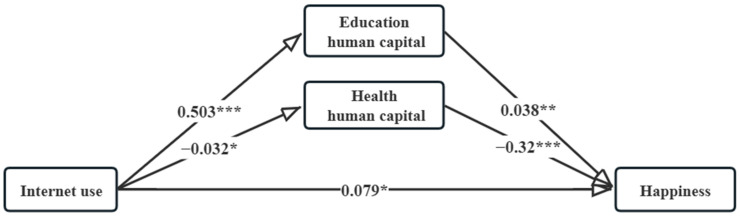
The Mediating Relationship Between Internet Use and Happiness. * *p* < 0.05, ** *p* < 0.01, *** *p* < 0.001.

**Figure 2 ijerph-20-03540-f002:**
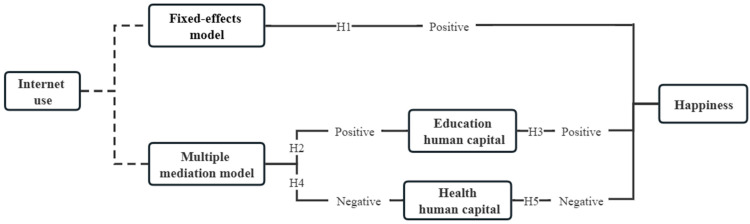
Test results of the theoretical and mechanism framework of the relationship between Internet use and rural residents’ happiness.

**Table 1 ijerph-20-03540-t001:** Descriptive statistics for regression model.

Category	Variables	Definition	Mean (SD)
Explanatory variables	Happiness	Self-rated happiness is 0–10	5.550 (2.508)
Explained variable	Internet use	Internet usage duration (min/day), take the logarithm	4.375 (1.006)
Control variables	Gender	Female = 0	48.32%
		Male = 1	51.68%
	Age	Years (take the logarithm)	3.421 (0.320)
	Education	Years of education	10.450 (3.239)
	Health	Self-rated health level 1–5	2.618 (1.062)
	Political landscape	yes = 1 And no = 0	0.074 (0.262)
	Family size	Number of family members	4.454 (1.905)
	Per capita household income	Household per capita income, take the logarithm	9.699 (0.877)
	Province	East = 1	42.88%
		Central = 2	29.70%
		West = 3	27.42%
	Social status	Respondents self-rated 1–5	2.807 (0.938)

**Table 2 ijerph-20-03540-t002:** Analysis of the impact of Internet use on rural residents’ happiness.

Variables	Happiness					
	Fe (1)	Re (2)	Fe (3)	Re (4)	Fe (5)	Re (6)
Internet use	0.070 *	0.241 ***	0.066 *	0.202 ***	0.063 *	0.207 ***
	(2.41)	(9.38)	(2.28)	(7.89)	(2.20)	(8.31)
Gender	0.512	−0.159 **	0.417	−0.163 **	0.525	−0.140 **
	(0.83)	(−3.12)	(0.67)	(−3.22)	(0.86)	(−2.83)
Age	56.071 ***	0.893 ***	55.215 ***	0.700 ***	53.978 ***	0.457 ***
	(94.04)	(10.07)	(87.45)	(7.81)	(85.66)	(5.19)
Years of education	−0.136 ***	0.071 ***	−0.130 ***	0.048 ***	−0.124 ***	0.045 ***
	(−4.28)	(8.30)	(−4.12)	(5.48)	(−3.98)	(5.27)
Health	−0.094 **	−0.282 ***	−0.095 **	−0.282 ***	−0.073*	−0.209 ***
	(−2.96)	(−11.44)	(−3.01)	(−11.50)	(−2.34)	(−8.71)
Political landscape	0.380	0.155	0.376	0.169	0.332	0.013
	(1.40)	(1.57)	(1.38)	(1.72)	(1.24)	(0.14)
Family size			0.082 **	0.080 ***	0.072 **	0.074 ***
			(3.15)	(5.63)	(2.82)	(5.36)
Per capita income			0.206 ***	0.414 ***	0.209 ***	0.412 ***
			(4.53)	(12.98)	(4.67)	(13.09)
Social status					0.385 ***	0.629 ***
					(12.14)	(23.78)
Region					Control	Control

Constant term	−185.187 ***	1.512 ***	−184.604 ***	−1.791 ***	−181.856 ***	−2.857 ***
	(−95.90)	(4.17)	(−94.22)	(−4.02)	(−93.31)	(−6.39)
Sample size	9482.000	9482.000	9482.000	9482.000	9482.000	9482.000
r2	0.701		0.703		0.712	
F-statistic	1852.486		1399.132		1064.470	
Prob	0.000	0.000	0.000	0.000	0.000	0.000

Standard errors in parentheses * *p* < 0.05, ** *p* < 0.01, *** *p* < 0.001.

**Table 3 ijerph-20-03540-t003:** Analysis of the mediating effects of household human capital.

Structure Model of Each Path	Coefficient	SD Coefficients	S.E.
Internet use → Education human capital	0.503 ***	0.182 ***	0.018
Internet use → Health human capital	−0.032 *	−0.041 *	0.018
Internet use → Happiness	0.079 *	0.042 *	0.019
Education human capital → Happiness	0.038 **	0.057 **	0.019
Health human capital → Happiness	−0.320 ***	−0.136 ***	0.018
Control variables (omitted)	Control		

Standard errors in parentheses * *p* < 0.05, ** *p* < 0.01, *** *p* < 0.001.

**Table 4 ijerph-20-03540-t004:** Analysis of the mediating effects of household human capital.

	Total Effect	Mediating Effects (1)	Mediating Effects (2)	Direct Effect
value	0.108 ***	0.019 **	0.010 *	0.079
95% confidence interval	[0.038, 0.175]	[0.007, 0.033]	[0.002, 0.021]	[0.009, 0.146]

Standard errors in parentheses * *p* < 0.05, ** *p* < 0.01, *** *p* < 0.001.

**Table 5 ijerph-20-03540-t005:** Heterogeneity analysis by region.

Structure Model of Each Path	Central	West	East
Internet use → Education human capital	0.4 ***	0.511 ***	0.423 ***
Internet use → Health human capital	−0.036	−0.028	−0.034
Internet use → Happiness	0.037	0.159 **	−0.032
Education human capital → Happiness	0.003	0.018	0.044 *
Health human capital → Happiness	−0.521 ***	−0.314 ***	−0.363 ***

Standard errors in parentheses * *p* < 0.05, ** *p* < 0.01, *** *p* < 0.001.

**Table 6 ijerph-20-03540-t006:** Heterogeneity analysis by family structure.

Structure Model of Each Path	(1)	(2)	(3)
Internet use → Education human capital	0.755 ***	0.568 ***	0.237 ***
Internet use → Health human capital	−0.078 *	−0.04	−0.002
Internet use → Happiness	−0.075	0.119 *	0.101 *
Education human capital → Happiness	0.079 ***	0.019	0.049 *
Health human capital → Happiness	−0.356 ***	−0.419 ***	−0.362 ***

Standard errors in parentheses * *p* < 0.05, *** *p* < 0.001.

## Data Availability

http://www.isss.pku.edu.cn/cfps/download (accessed on 25 January 2022).

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
