# Peer review of "Internet Use and the Happiness of Rural Residents: The Role of Education and Health"

_ijerph, 2023, doi:10.3390/ijerph20043540_

Round 1

Reviewer 1 Report

It´s a good job, just in the results, I recommend can be clearer to understand them.

Reviewer 2 Report

Dear authors,

Congratulations on a wonderfully written and interesting article. The amount of thought and effort put in this manuscript is apparent. Apart from some minor suggestions listed below, I believe that this manuscript could be published.

The introduction is long. Several parts can be moved to the literature review section (Lines 78-98). Also please highlight the scope of research in a separate paragraph. It would be useful to present the research questions in a bulleted fashion.

Furthermore, the hypotheses tested in this article should be presented at the end of the introduction following the scope of research (Lines 158-214).

Line 262: Please fix the second “very healthy” scale

A small section dedicated to future research would be useful. The impact of internet on happiness during the pandemic would also be an interesting read.
